# Ecological Niche Modeling of *Aedes* and *Culex* Mosquitoes: A Risk Map for Chikungunya and West Nile Viruses in Zambia

**DOI:** 10.3390/v15091900

**Published:** 2023-09-08

**Authors:** Rachel Milomba Velu, Geoffrey Kwenda, Samuel Bosomprah, Moses Ngongo Chisola, Michelo Simunyandi, Caroline Cleopatra Chisenga, Flavien Nsoni Bumbangi, Nicholus Chintu Sande, Limonty Simubali, Monicah Mirai Mburu, John Tembo, Matthew Bates, Martin Chitolongo Simuunza, Roma Chilengi, Yasuko Orba, Hirofumi Sawa, Edgar Simulundu

**Affiliations:** 1Centre for Infectious Disease Research in Zambia, Lusaka P.O. Box 34681, Zambia; samuel.bosomprah@cidrz.org (S.B.); michelo.simuyandi@cidrz.org (M.S.); caroline.chisenga@cidrz.org (C.C.C.); roma.chilengi@cidrz.org (R.C.); 2Department of Disease Control, School of Veterinary Medicine, University of Zambia, Lusaka P.O. Box 32379, Zambia; martin.simuunza@unza.zm (M.C.S.); h-sawa@czc.hokudai.ac.jp (H.S.); 3Department of Biomedical Sciences, School of Health Sciences, University of Zambia, Lusaka P.O. Box 50110, Zambia; kwenda.geoffrey@unza.zm; 4Department of Biostatistics, School of Public Health, University of Ghana, Accra P.O. Box LG13, Ghana; 5Department of Geography and Environmental Studies, School of Natural Sciences, University of Zambia, Lusaka P.O. Box 32379, Zambia; mchisola@yahoo.com; 6Department of Medicine and Clinical Sciences, School of Medicine, Eden University, Lusaka P.O. Box 37727, Zambia; bnflavien@gmail.com; 7National Malaria Elimination Centre, Chainama Hills Hospital Grounds, Lusaka P.O. Box 32509, Zambia; nicholus.sande@gmail.com; 8Macha Research Trust, Choma P.O. Box 630166, Zambia; limonty.simubali@macharesearch.org (L.S.); monicah.mburu@macharesearch.org (M.M.M.); 9HerpeZ, University Teaching Hospital, Lusaka 10101, Zambia; john.tembo@gmail.com (J.T.); mbates@lincoln.ac.uk (M.B.); 10Joseph Banks Laboratories, School of Life and Environmental Sciences, University of Lincoln, Lincolnshire LN6 7TS, UK; 11Africa Centre of Excellence for Infectious Diseases of Humans and Animals, University of Zambia, Lusaka P.O. Box 32379, Zambia; 12Zambia National Public Health Institute, Ministry of Health, Lusaka P.O. Box 51925, Zambia; 13Division of Molecular Pathobiology, International Institute for Zoonosis Control, Hokkaido University, N 20 W10, Kita-Ku, Sapporo 001-0020, Japan; orbay@czc.hokudai.ac.jp; 14International Collaboration Unit, International Institute for Zoonosis Control, Hokkaido University, Hokkaido 060-0808, Japan; 15One Health Research Center, Hokkaido University, Sapporo 001-0020, Japan; 16Institute for Vaccine Research and Development, Hokkaido University, Sapporo 001-0021, Japan; 17International Collaboration Unit, Global Virus Network, Baltimore, MD 21201, USA

**Keywords:** arbovirus, *Aedes*, *Culex*, West Nile virus, Chikungunya virus, ecological niche modeling, Zambia

## Abstract

The circulation of both West Nile Virus (WNV) and Chikungunya Virus (CHIKV) in humans and animals, coupled with a favorable tropical climate for mosquito proliferation in Zambia, call for the need for a better understanding of the ecological and epidemiological factors that govern their transmission dynamics in this region. This study aimed to examine the contribution of climatic variables to the distribution of *Culex* and *Aedes* mosquito species, which are potential vectors of CHIKV, WNV, and other arboviruses of public-health concern. Mosquitoes collected from Lusaka as well as from the Central and Southern provinces of Zambia were sorted by species within the Culex and *Aedes* genera, both of which have the potential to transmit viruses. The MaxEnt software was utilized to predict areas at risk of WNV and CHIKV based on the occurrence data on mosquitoes and environmental covariates. The model predictions show three distinct spatial hotspots, ranging from the high-probability regions to the medium- and low-probability regions. Regions along Lake Kariba, the Kafue River, and the Luangwa Rivers, as well as along the Mumbwa, Chibombo, Kapiri Mposhi, and Mpika districts were predicted to be suitable habitats for both species. The rainfall and temperature extremes were the most contributing variables in the predictive models.

## 1. Introduction

Emerging and re-emerging arboviral infectious diseases, such as Chikungunya Virus (CHIKV) and West Nile Virus (WNV), are a significant threat to global-health security, particularly in sub-Saharan Africa where large epidemics have been observed [1,2]. CHIKV is an alphavirus transmitted by *Aedes aegypti* and *Aedes albopictus* mosquitoes [3,4]. It has caused several outbreaks globally and its symptomatology is characterized by an acute weakening syndrome, often accompanied by fever, polyarthralgia, and rash [5,6]. CHIKV is endemic in various African countries, including Uganda, the Democratic Republic of Congo, Zimbabwe, Senegal, Nigeria, South Africa, Tanzania, and Kenya [3]. On the other hand, WNV belongs to the Flavivirus genus in the Flaviviridae family and is a member of the Japanese encephalitis (JE) virus serocomplex transmitted by *Culex* mosquitoes [7,8]. WNV infects humans, equines, domestic animals, and wild birds [7]. The transmission of WNV occurs through an enzootic cycle, where birds serve as vertebrate hosts and mosquitoes act as the primary vectors [9]. Most WNV infections in humans are asymptomatic, but in some cases, flu-like symptoms, fever, myalgia, and severe neurological complications, such as meningoencephalitis and flaccid paralysis, may occur, leading to death [7,10]. 

The occurrence of mosquito-borne viruses, including CHIKV and WNV in Zambia, has been investigated mainly through human serological studies, which found the circulation of CHIKV in the Central province at a prevalence of 36.9 percent and WNV in the Northwestern and Western provinces at a prevalence of 10.3 percent [11,12,13,14,15]. Furthermore, WNV was detected in Culex mosquitoes in the Western province and in farmed crocodiles in the Southern province [16,17]. Despite evidence of the occurrence of CHIKV and WNV in the country, there is no routine surveillance of these mosquito-borne viruses. Although Zambia has not reported any epidemics of WNV or CHIKV in the last 3 decades, the favorable tropical climate for mosquito breeding and the occurrence of outbreaks in neighboring countries [5,18,19] warrants an understanding of the ecological factors influencing the dynamics of arboviral transmission and their natural maintenance cycles in the environment. To achieve this understanding, Ecological Niche Modeling (ENM), also known as Species Distribution Modeling (SDM), has been employed worldwide to study arboviruses such as CHIKV, Yellow Fever Virus (YFV), and WNV, and their vectors [20,21,22,23,24]. 

This study aimed to examine the contribution of climatic variables to the distribution of *Culex* and *Aedes* mosquito species, which are potential vectors of WNV and CHIKV, respectively. This will help in implementing appropriate and targeted surveillance strategies and epidemic preparedness for the country. 

## 2. Materials and Methods

### 2.1. Study Areas

Zambia, located in the southern part of Africa between latitudes 8° and 18° south and longitudes 22° and 34° east, covers a total area of 752, 618 square kilometers, of which 9220 km^2^ are covered by water. The country experiences three distinct seasons: the cool-dry season from May to August, the hot-dry season from August to November, and the hot-wet season from November to April [25]. Zambia is divided into three agroecological zones, including Region I, which is the driest zone, with a mean annual rainfall of less than 800 mm, and this zone covers the Lusaka Province, the Southern Province, and a part of the Eastern and Western provinces. Region II receives between 800 and 1000 mm of rain and Region III has a mean annual rainfall of 1000 to 1500 mm [26]. Zone II covers the Western and Central provinces, as well as a part of the Eastern and Southern provinces. The study sites included Lusaka, Sinazongwe, and the Kapiri districts in Lusaka and in the Southern and Central provinces, respectively, which are part of Regions I and II. These sites were chosen based on previous reports of mosquito-borne viral activity in the country [27].

### 2.2. Mosquito Occurrence Data

Adult mosquitoes (*n* = 834) were trapped within the Lusaka, Kapiri, and Sinazongwe districts in February and May 2021 using the CO_2_-baited CDC light traps (John W. Hock Co., Gainesville, FL, USA). The traps were set indoors and outdoors from the afternoon (15:00 h) to the next morning (09:00 h). Outdoor traps were placed closer to potential mosquito-breeding habitats, whereas indoor traps were placed in bedrooms closer to the bed during the sleeping hours. The sampling was conducted for 3 nights at each site. Captured mosquitoes were frozen at −20 °C and sorted out per sex and species on ice packs using the morphological referencing keys of African mosquitoes [28]. After sorting, a total of 12 mosquito-field occurrence points were recorded after removing spatially autocorrelated records. An additional 37 occurrence points of both *Culex* and *Aedes* were obtained from published literature [16,29,30] (Appendix A). 

### 2.3. Environmental Covariates

Nineteen (19) suitable environmental covariates and one altitude variable were retrieved from WorldClim [31] at a spatial resolution of 10 km. The environmental covariates incorporated in the models included temperature and its extremes: annual mean temperature, mean diurnal range, temperature seasonality, maximum temperature of the warmest month, minimum temperature of the coldest month, temperature annual range, mean temperature of the wettest quarter, mean temperature of the driest quarter, mean temperature of the warmest quarter, and mean temperature of the coldest quarter. Others were moisture variables (annual precipitation, precipitation of the wettest month, precipitation of the driest month, precipitation seasonality, precipitation of the wettest quarter, precipitation of the driest quarter, precipitation of the warmest quarter, and precipitation of the coldest quarter). We used ENMTOOLs to test for multicollinearity between the predictor variables [32]. We ran a pairwise Pearson correlation and only variables with a correlation of less than (±0.75) were retained in the final prediction models (Appendix A). After accounting for multicollinearity, only nine variables comprising three moisture variables, five temperature variables, and one altitude variable were retained (Table 1). 

#### Modeling Procedure

For the ecological modeling, we used MaxEnt version 3.3 k, with the settings determined by the ENMEval R package in RStudio. Based on the lowest delta AIC model for the *Culex* species, a linear features model with a regularization multiplier of 1 and 10,000 background points was needed. For the *Aedes* species, the settings were the same, but with a regularization multiplier of 2. These model settings were then used to run the suitability models in MaxEnt. Seventy-five percent of the occurrence data was randomly set aside for training the model and 25% for testing or validating the accuracy of the model. The independent variables input into MaxEnt were the spatial layers of the 9 environmental variables in ASCII-file format. The dependent variable was the species’ (*Aedes* and *Culex*) presence-point coordinates in a CSV-file format. The predictive performance of the model was assessed using the Receiver Operating Curve (ROC) [20]. The MaxEnt outputs were imported into the QGIS 2.18.3 mapping software to develop the final maps. The contribution of each environmental variable to the models was tested using the Jackknife method built-in to MaxEnt. The Jackknife analysis provides three types of model outputs: (1) a full model created with all variables; (2) several models created with all variables, but with one variable excluded at a time; and (3) several models created with one variable at a time. Variables with high contribution to the model outputs were those showing the biggest Area Under the Curve (AUC) when included separately and the smallest AUC when removed [33]. The distribution maps were generated in MaxEnt from 50 bootstrap replicates.

## 3. Results

### 3.1. Distribution of Mosquitoes across the Sampling Areas

A total of 834 adult mosquitoes belonging to three genera were captured in three of the ten provinces of Zambia. *Culex* (*n* = 652) mosquitoes were the most predominant across the three provinces. A total of 87 *Aedes* were captured across the three provinces. The other genus was *Anopheles* (not included in this study as they are not potential vectors of both viruses under study). Figure 1 illustrates the distribution of the occurrence data in the study areas as well as from the literature.

### 3.2. Model Performance

The MaxEnt model outputs provided AUC values of 0.761 and 0.821 for both *Aedes* and *Culex*, respectively, indicating an acceptable model performance (Figure 2). The test revealed a statistically significant difference between the AUC from the model prediction and the AUC at a randomness set at 0.5 [20].

### 3.3. Vectors’ Habitat Suitability 

Overall, the MaxEnt model outputs predicted suitable vector habitats with high accuracy rates. We found that the precipitation seasonality (BIO18; 85%), precipitation of the driest quarter (BIO17; 14.3%), precipitation of the coldest quarter (BIO19; 0.4%), and temperature seasonality (BIO4; 0.2%) had the highest contributions to the *Aedes* species model distribution (Table 2 and Figure 3A). However, in the *Culex* model outputs, the precipitation of the warmest quarter (BIO18; 41.2%), precipitation of the driest quarter (BIO17; 37.7%), temperature seasonality (BIO4; 11.7%), precipitation of the coldest quarter (BIO19; 3.7%), precipitation of the wettest quarter (BIO16; 3.0%), elevation (1.2%), and precipitation seasonality (BIO15; 1.0%) significantly influenced the *Culex* species’ suitability habitat (Table 2 and Figure 3B).

The Relative Probability of Presence (RPP) ranged from very suitable areas with a probability value of closer to 0.98% (Figure 4A). The findings show that most parts of Zambia are suitable for *Aedes* breeding but most notably in the following districts: Kapiri-Mposhi, Kabwe, Chisamba, and Chibombo (in the Central province); Mazabuka, Chikankata, Chirundu, Siavonga, Gwembe, Monze, and Kazungula (in the Southern province); Lusaka, Chongwe, Chilanga, Kafue, and Luangwa (in the Lusaka province); Sesheke, Mwandi, Sioma, and Luampa (in the Western province); Kanchibiya (in the Muchinga province); Chipanguli (in the Eastern province); and Chinsali (in the Northern province). Moderately suitable regions cover most districts of the Northwestern and Northern provinces (Figure 4A).

High RPP for the *Culex* species was predicted in most districts in the following provinces: Central (Kapiri-Mposhi, Kabwe, and Chibombo), Lusaka (Lusaka, Chongwe, and Chilanga), Southern (Mazabuka and Chikankata), Eastern (Chipanguli and Kasenengwa), Muchinga (Mpika), and Western (Kaoma, Nkeyema, Luampa, and Sioma). Moderately suitable regions cover most parts and districts of the Northwestern, Western, and Copperbelt provinces (Figure 4B). The RPP of *Culex* was mainly influenced by the precipitation of the warmest quarter at 41.2%, precipitation of the driest quarter at 37.7%, temperature seasonality at 11.7%, and precipitation of the coldest quarter at 3.7% (Table 2).

The response curves of the variables that contributed the most to the *Aedes* and *Culex* species’ habitat suitability are shown in Figure 5 and Figure 6. Precipitation of less than 100 mm during the warmest quarter of the year (Figure 5A) increased the probability of *Aedes* species’ occurrence to closer to 99%; however, this probability quickly dropped as the precipitation increased to above 300 mm, reaching a probability of 30%. This was expected as high-intensity rainfall would probably destroy both larvae and, in some instances, adult mosquitoes. The probability of *Aedes* occurrence is at a maximum with the first rainfall during the driest quarter of the year (Figure 5B) and it decreases with increasing rainfall amounts. Furthermore, it varies with the change in temperature over the course of the year. Increased temperature variations were related to increasing likelihoods of *Aedes* species’ occurrence. The optimal occurrence would take place when the temperature variations from the mean are above 25 °C (Figure 5C).

On the other hand, the model outputs for the *Culex* species showed that their probability of occurrence is optimal at minimum rainfalls (110 mm) during the warmest quarter of the year and drops until the flattening of the curve when precipitation is above 200 mm (Figure 6A). Furthermore, the suitability of the presence of the *Culex* species increases when the temperature is closer to 27 °C, reaching a probability of 80% and dropping very quickly to closer to 20% with a temperature above 27 °C (Figure 6B). Furthermore, elevation greatly affected the distribution of the *Culex* species as the probability of finding the *Culex* species increases with an altitude above 2000 m (Figure 6C). Precipitation of the wettest quarter (BIO 16) is an index that provides the total precipitation during the wettest three months of the year (Figure 6D). The probability of occurrence of Culex mosquitoes increases with the first rainfalls during this quarter, reaches a peak at closer to 590 mm of precipitation with an 82% probability, and soon reduces at 42% for an increased amount of rainfall (above 620 mm of precipitation; Figure 6D).

## 4. Discussion

In this study, occurrence records of the *Aedes* and *Culex* species from different provinces of Zambia were used as presence-only data to study their environmental suitability. The knowledge gained on their current distribution is of great value in identifying the areas at risk of CHIKV, WNV, and other arboviruses transmitted by both mosquito species for targeted surveillance.

Our study revealed the occurrence of both the *Aedes* and *Culex* species in the sampling areas during the study period, with the *Culex* species appearing in larger numbers. The low number of the *Aedes* species in the sampling areas could be attributed to the collection time as traps were set in the afternoon around 15:00 (h) whilst *Aedes* mosquitos are day-time feeders. Furthermore, for both species, the short duration of the sampling could have, as well, played a role in their abundance. Nonetheless, their presence, even in small populations, calls for monitoring and an investigation of the potential risk of arboviral disease transmission.

Overall, the model predictions for the distribution of the *Culex* and *Aedes* mosquito species showed three distinct spatial hotspots ranging from the regions of high to moderate and low probability of occurrence. Regions of high RPP for both vectors mostly corresponded to areas that reflect specific environmental conditions for each species, such as rainfall patterns. The important environmental predictors for the final environmental suitability models were mainly related to precipitation and temperature extremes. 

Even though the training AUC values for both the *Aedes* and *Culex* species could be considered low (0.76 and 0.82, respectively) compared to other studies that found AUC values above 90% for the same species [21,22], our predictions remain above the randomness’ prediction value set at 50% [20]. The difference in terms of AUC values might be due to the number of occurrence points used in the models. Our results, however, agree with those from various authors [23,24,34] in terms of AUC values.

The association between mosquitoes and precipitation is not surprising as the life cycle of mosquitoes at larval and pupae stages requires water [35]. These results are consistent with the findings of various authors who explored the impact of precipitation variability on mosquito development and abundance [30,35]. On the other hand, a minimum amount of precipitation during the warmest quarter, in the case of Zambia, corresponding to mid-August to mid-November, would have a positive impact on the distribution of both mosquito species. This could be explained by the rise in temperature observed in that quarter of the year. However, the response of the mosquitoes to the variation of temperature is species-specific; some species survive in higher temperatures (e.g., *Culex quinquefasciatus*), whereas others (e.g., *Culex pipiens*) may be more sensitive [36]. Furthermore, it was noted that temperature variations between 25 °C and 27 °C would increase the probability of occurrence of both species and higher temperatures (above 27 °C) would be a limiting factor to their occurrence. These findings agree with several authors who explored the effect of temperature on mosquitoes’ development [37,38,39].

In the *Culex* model outputs, elevation was positively associated with the probability of occurrence. This result disagrees with previous findings that observed that highly elevated areas were unsuitable for *Culex* species’ development [40,41]. This is unsurprising because in an ideal scenario, the higher the elevation, the cooler it becomes, and this should normally affect the development and survival capacity of the *Culex* species; however, climate warming or any other factor that modifies the microclimatic conditions, such as land use, land cover, and deforestation, could likely shift and promote their presence and distribution even in areas previously unsuitable, such as high-elevation areas, as it has been reported with other mosquitoes’ species, such as *Anopheles* and *Aedes* [42,43,44,45]. Interestingly, Carolina Garcia et al. (2010) found that some *Culex* species, specifically *Culex quinquefasciatus*, had been adapting to higher-elevation zones [46]. Furthermore, a study conducted in four high-altitude areas in East Africa using temperature data from 1950 to 2002 revealed a noticeable warming trend at all four study sites and projected that a 0.5 °C rise in temperature may result in an increase of 30–100% in mosquito abundance [47]. 

The *Culex* and *Aedes* probability maps showed that most parts of the country are suitable habitats for these species but mainly the regions along Lake Kariba, the Kafue River, and the Luangwa river-basin districts, as well as the Mumbwa, Chibombo, Kapiri, and Mpika districts. Interestingly, most of these areas have reported some mosquito-borne viral activity, which was confirmed by serological surveys and the detection of viruses [11,17,27,30,48], except for the districts in the Muchinga province; this exception may imply a lack of documented data rather than an actual absence of the circulation of mosquito-borne viruses given the favorable environmental conditions in the area.

It is worth highlighting that vector-distribution models reflect a potential distribution of vector-borne pathogens rather than their complete distribution [33]. Nevertheless, modeling the vectors is, foremost, more effective than modeling their hosts, specifically human hosts as these are subject to high-dispersal ranges due to their mobility, for example.

Despite using few occurrence points for the mosquito species, MaxEnt has shown to be successful in generating biologically meaningful models with minimal occurrence records of as few as six [49]. Furthermore, we used mostly climatic or environmental variables as key drivers of suitability and/or the geographic distribution of the studied species; this should be taken with caution as the suitability habitats were modeled with the assumption that the species will not encounter any dispersed limitation. Our prediction is an ideal state and should be considered as a point of reference for targeted surveillance in areas identified as high-risk for mosquito-borne viruses.

Although Zambia has not reported any epidemics of WNV or CHIKV in the last three decades, the tropical climate of Zambia offers favorable breeding sites for mosquitoes that can transmit both WNV and CHIKV. Furthermore, the continuous mosquito-borne viral-disease outbreaks in the neighboring countries, such as in Angola and the Democratic Republic of Congo, places Zambia at risk of mosquito-borne outbreaks. Furthermore, the absence of epidemics does not mean that the viruses are not circulating in the country, as evidenced by the serological studies conducted by other scholars. Thus, there is a need to alert decision-making officials to strengthen surveillance in the country, especially in the hotspot points outlined in this manuscript.

## 5. Conclusions

In this study, we used the occurrence records of mosquitoes (*Aedes* and *Culex*) in conjunction with environmental covariates to assess their habitat suitability and/or distribution. Overall, the model predictions show three distinct spatial hotspots, ranging from the high-probability regions to the medium- and low-probability regions. The precipitation of the warmest and driest quarters as well as the temperature seasonality were important predictors of the *Culex* and *Aedes* species, which are potential vectors of WNV, CHIKV, and other arboviruses of public-health importance. The probability maps showed a wide range of areas that could be potential mosquito-borne infectious-disease hotspots with climate-change phenomena, which could, consequently, lead to emerging diseases. These models can be used by public-health officials under the “One Health” umbrella for targeted arboviral surveillance primarily in high-probability areas in the country.

## Figures and Tables

**Figure 1 viruses-15-01900-f001:**
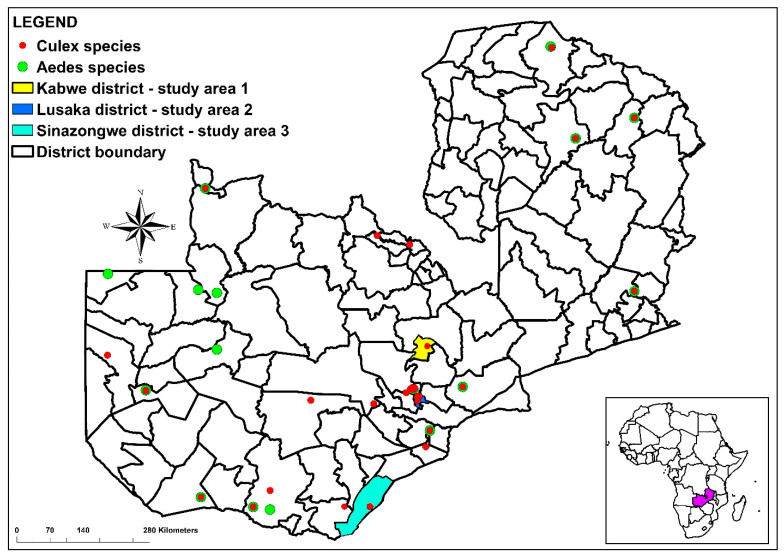
Map of Zambia showing locations of *Culex* and *Aedes* occurrence records included in MaxEnt modeling. The red and green dots represent the *Culex* and Aedes species, respectively. The overlap of red and green points indicates that both species were captured in the location. The yellow, dark blue, and light blue colors represent the sampling areas.

**Figure 2 viruses-15-01900-f002:**
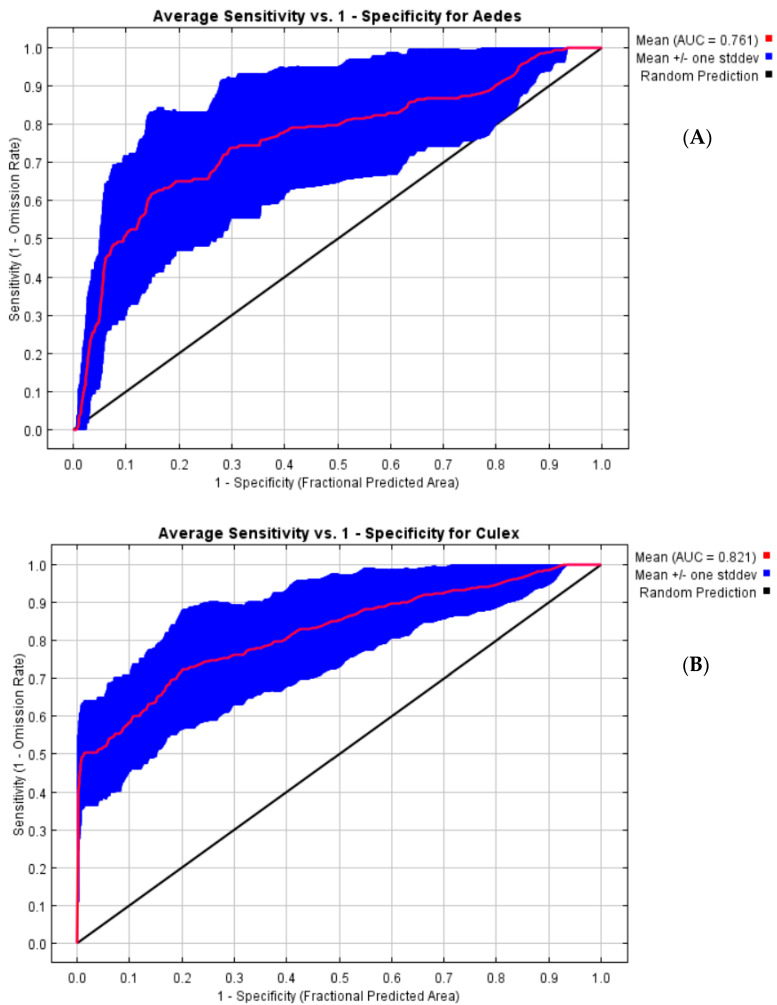
ROC curves for MaxEnt *Aedes* (**A**) and *Culex* (**B**) models’ performance (red line: training and testing data; blue line: standard deviation; and black line: random prediction). The AUC had values of 0.761 and 0.821 for *Aedes* and *Culex*, respectively.

**Figure 3 viruses-15-01900-f003:**
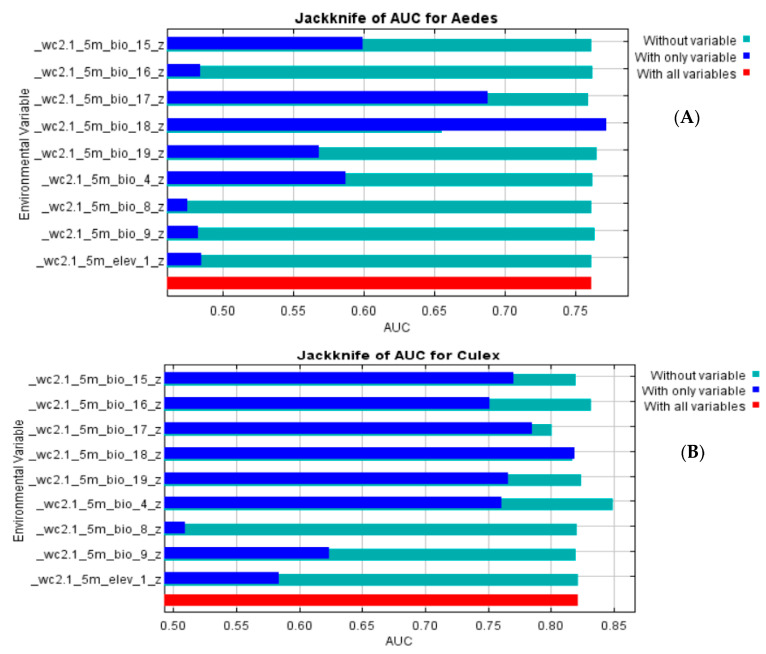
Jackknife of test gain for *Aedes* (**A**) and *Culex* (**B**). The blue bars show the effect of each variable on the model by itself, while the light blue bars show the effect in the model when this variable is not considered. The red bar represents the performance of the model when all variables are included in the model.

**Figure 4 viruses-15-01900-f004:**
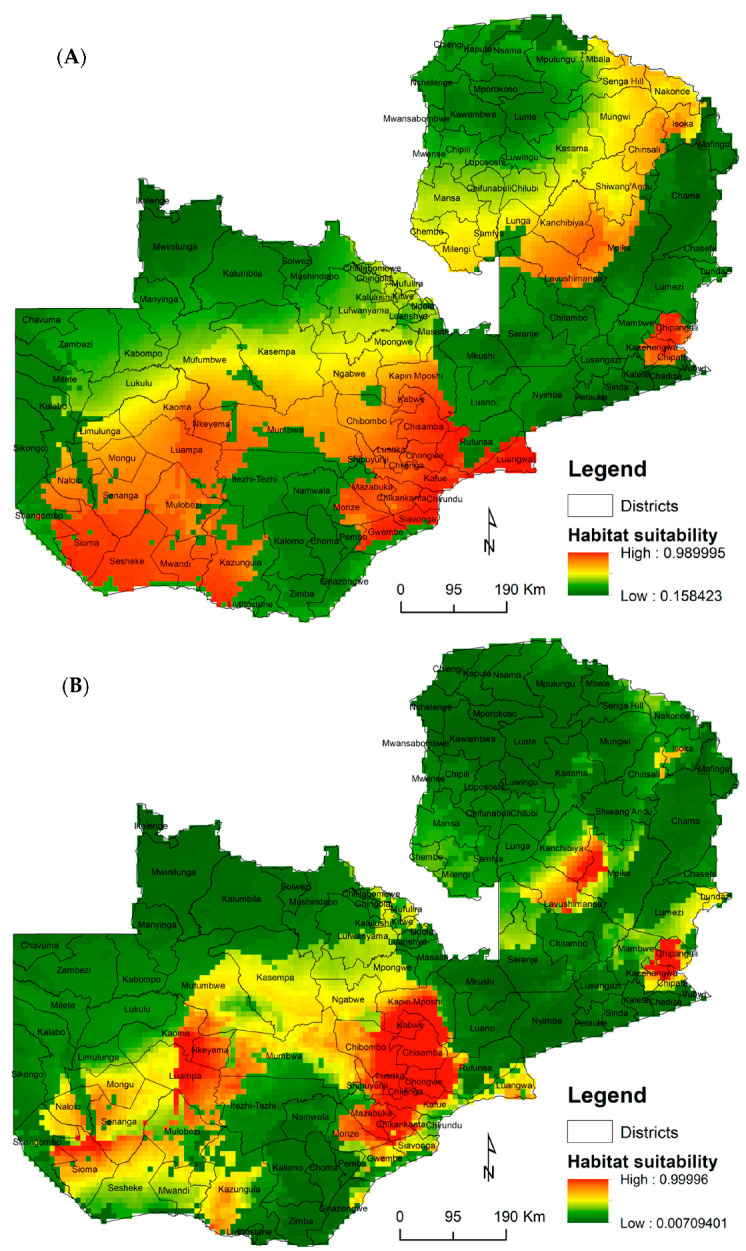
Potential probability distribution of *Aedes* (**A**) and *Culex* (**B**) species in Zambia. Areas of high suitability are represented by the orange color through the light yellow color and less suitable areas are represented by the green color.

**Figure 5 viruses-15-01900-f005:**
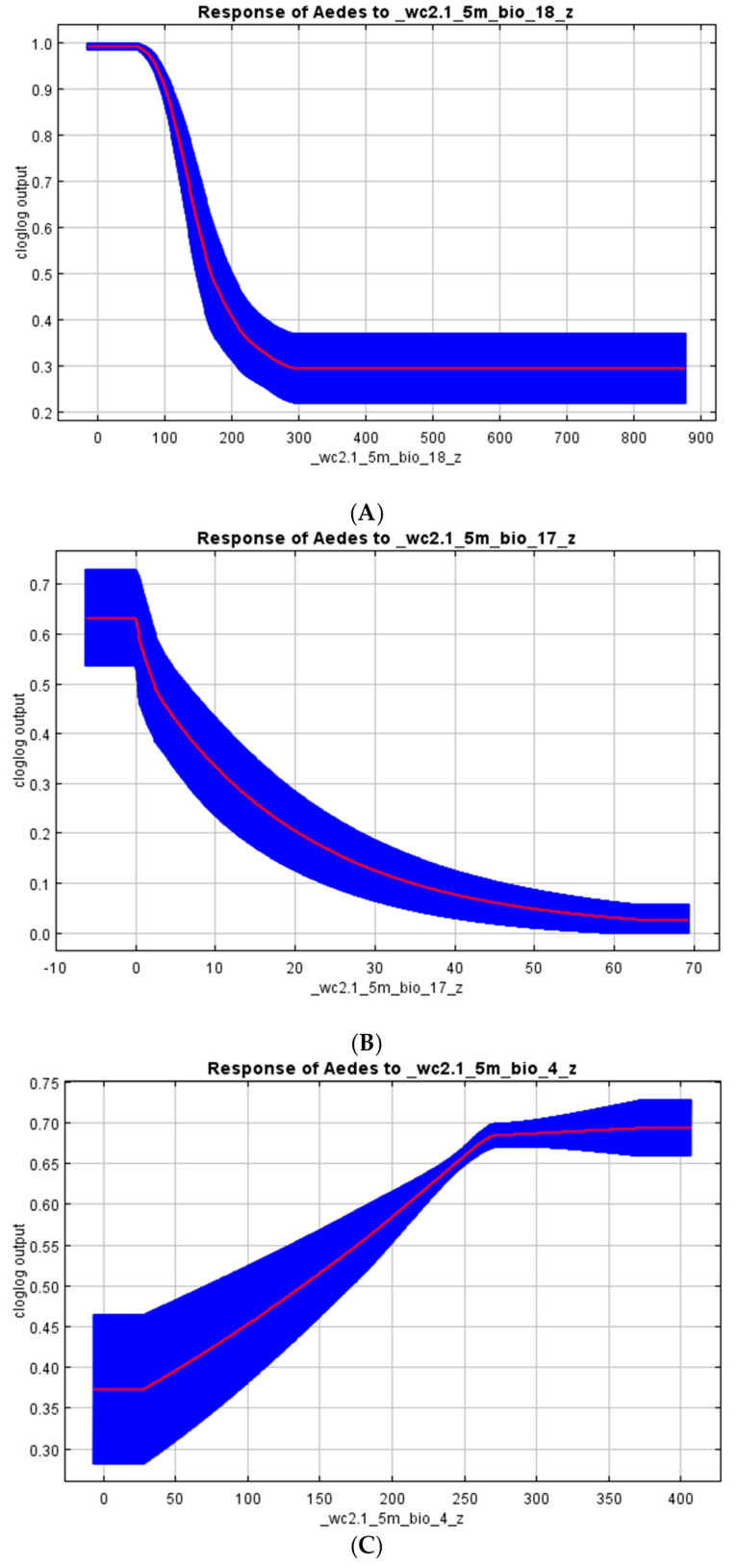
The effect of each environmental variable that most contributes to the suitability habitat of Aedes species when the remaining variables are configured to their average value. (**A**): BIO 18, precipitation of the warmest quarter, for which the units are millimeters of precipitation; (**B**): BIO 17, precipitation of the driest quarter, for which the units are millimeters of precipitation; and (**C**): BIO 4, temperature seasonality, for which the units are degrees Celsius by 10.

**Figure 6 viruses-15-01900-f006:**
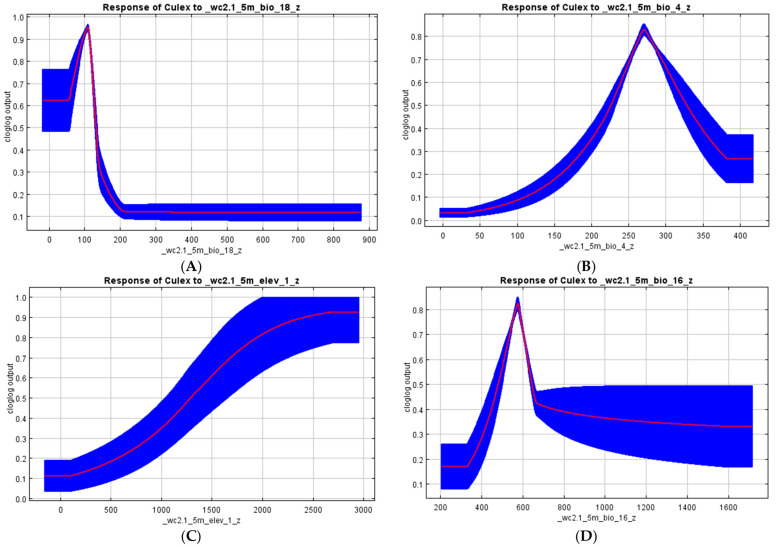
Shows the effect of each environmental variable that most contributes to the suitability habitat of *Culex* species when the remaining variables are configured to their average value. (**A**): BIO 18, precipitation of the warmest quarter, for which the units are millimeters of precipitation; (**B**): BIO 4, temperature seasonality, for which the units are degrees Celsius by 10; (**C**): elevation, for which X-axis units are millimeters above sea level; and (**D**): BIO 16, precipitation of wettest quarter (BIO 16), for which the units are millimeters of precipitation.

**Table 1 viruses-15-01900-t001:** Environmental variables used in the models (adapted from WorldClim.org).

Variable Description	Abbreviation	Unit
Temperature seasonality (standard deviation × 100)	BIO4	°C
Mean temperature of wettest quarter	BIO8	°C
Mean temperature of driest quarter	BIO9	°C
Precipitation seasonality (coefficient of variation)	BIO15	Mm
Precipitation of wettest quarter	BIO16	Mm
Precipitation of driest quarter	BIO17	Mm
Precipitation of warmest quarter	BIO18	Mm
Precipitation of coldest quarter	BIO19	Mm
Elevation	N/A	M

**Table 2 viruses-15-01900-t002:** Percent contribution of environmental variables for final models (*Aedes* and *Culex* species).

*Culex* Model		*Aedes* Model	
Environmental Variable	Contribution (%)	Environmental Variable	Contribution (%)
Precipitation of warmest quarter (BIO18)	41.2	Precipitation of warmest quarter (BIO18)	85
Precipitation of driest quarter (BIO17)	37.7	Precipitation of driest quarter (BIO17)	14.3
Temperature seasonality (standard deviation × 100; BIO4)	11.7	Precipitation of coldest quarter (BIO19)	0.4
Precipitation of coldest quarter (BIO19)	3.7	Temperature seasonality (standard deviation × 100; BIO4)	0.2
Precipitation of wettest quarter (BIO16)	3	Mean temperature of driest quarter (BIO9)	0
Elevation	1.2	Precipitation of wettest quarter (BIO16)	0
Precipitation seasonality (coefficient of variation; BIO15)	1	Elevation	0
Mean temperature of driest quarter (BIO9)	0.4	Mean temperature of driest quarter (BIO8)	0
Mean temperature of driest quarter (BIO8)	0.1	Precipitation seasonality (coefficient of variation; BIO15)	0

## Data Availability

All relevant data are within the paper and its Appendix A.

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
