# Peer review of "Ecological Niche Modeling of Aedes and Culex Mosquitoes: A Risk Map for Chikungunya and West Nile Viruses in Zambia"

_viruses, 2023, doi:10.3390/v15091900_

Round 1

Reviewer 1 Report

The manuscript titled "Ecological Niche Modelling of Aedes and Culex Mosquitoes: A Risk Map for Chikungunya and West Nile Viruses in Zambia" has been written very well and result and conclusion support each other. It is a very good study to find the co-relation between vector borne disease and vector availability. 

Minor point

1. As author mentioned that Zambia has not reported any epidemics of WNV or CHIKV in the last 3 decades, then how this study will be useful for Zambia in near future. 

2. In figure 1, what is in the insert. Please mention it in the figure legend. 

Reviewer 2 Report

The manuscript presents some very interesting results regarding ecological aspect of Aedes and culex Distribution. The Authors may implement Introduction with more information about surveillance in Zambia and in Materials and methods about Mosquito sampling.

Introduction

The Authors could add some information regarding serological surveillance of WN and CHIK in Zambia. 

Line 85:  invert order of abbreviations

Materials and methods

in Mosquitoes occurance data the Authors could add some information regarding distance to different sites.

line 105-112: Specify further how the choice of sites came about what were the variables that influenced the choice of where to place the traps?
